# Association of Hallux Valgus with Degenerative Spinal Diseases: A Population-Based Cohort Study

**DOI:** 10.3390/ijerph20021152

**Published:** 2023-01-09

**Authors:** Ta-Li Hsu, Yung-Heng Lee, Yu-Hsun Wang, Renin Chang, James Cheng-Chung Wei

**Affiliations:** 1Department of Orthopedic Surgery, Far Eastern Memorial Hospital, New Taipei City 220, Taiwan; 2Department of Senior Services Industry Management, Minghsin University of Science and Technology, Hsinchu 304, Taiwan; 3Department of Recreation and Sport Management, Shu-Te University, Kaohsiung 824, Taiwan; 4Department of Orthopedics, Cishan Hospital, Ministry of Health and Welfare, Kaohsiung 842, Taiwan; 5Department of Medical Research, Chung Shan Medical University Hospital, Taichung 402, Taiwan; 6Department of Emergency Medicine, Kaohsiung Veterans General Hospital, Kaohsiung 813, Taiwan; 7Department of Recreation Sports Management, Tajen University, Pingtung 907, Taiwan; 8Institute of Medicine, Chung Shan Medical University, Taichung 402, Taiwan; 9Department of Allergy, Immunology & Rheumatology, Chung Shan Medical University Hospital, Taichung 402, Taiwan; 10Graduate Institute of Integrated Medicine, China Medical University, Taichung 404, Taiwan

**Keywords:** hallux valgus, spondylosis, cohort, epidemiology

## Abstract

Background: Although hallux valgus is known to cause lower-back pain, the association between hallux valgus and spinal degenerative disease remains unclear. Methods: A retrospective cohort study was conducted between 1 January 2000 and 31 December 2015 using data from the Longitudinal Health Insurance Database in Taiwan. After propensity score matching for age, sex, and some potential comorbidities, 1000 individuals newly diagnosed with hallux valgus were enrolled in the study group, while 1000 individuals never diagnosed with hallux valgus served as the control group. Both groups were followed up until 2015 to evaluate the incidence of hallux valgus. Kaplan-Meier analysis was used to determine the cumulative incidence of hallux valgus, while the Cox proportional hazard model was adopted to estimate the hazard ratio (HR) and adjusted hazard ratio (aHR) with 95% confidence intervals (CIs). Results: The incidence densities of spinal degeneration in the hallux valgus and non-hallux valgus groups were 73.10 and 42.63 per 1000 person-years, respectively. An increased risk of spinal degenerative changes was associated with hallux valgus (adjusted HR = 1.75, 95% CI = 1.50–2.05). Age- and sex-stratified analyses showed a significantly higher risk of spinal degeneration in the hallux valgus group. Moreover, sub-outcome evaluations revealed significantly higher risks of spondylosis (aHR = 2.01, 95% CI = 1.55–2.61), intervertebral disorder (aHR = 2.27, 95% CI = 1.62–3.17), and spinal stenosis (aHR = 1.24, 95% CI = 1.47–1.76). There was also an increased risk of spinal degenerative change in those with hallux valgus without surgical intervention (aHR = 1.95, 95% CI = 1.66–2.99, *p* < 0.001). Conclusions: Hallux valgus was associated with increased risk of degenerative spinal changes and other spinal disorders.

## 1. Introduction

The spine plays an important role in supporting the trunk and protecting the neural system. Multidirectional motions and absorption of axial loading are known contributors to degenerative changes in the functional spinal unit [1]. Each functional spinal unit, which consists of a pair of symmetrical zygapophysial joints and a flexible intervertebral disc, is considered the fundamental structure of the spine [1,2]. The stability of the spinal column is reinforced by the anterior and posterior longitudinal ligaments, the ligamentum flavum, muscles, physiological curves, the vertebral architecture, and bone mineral density [3].

The degenerative changes of the spine are thought to be a dynamic process that involves the nucleous pulposus, annulus fibrosusm end plates, bone marrow, facet joints, ligamentum flavum, and spinal canal. On the basis of changes in MRI signals, Pfirrmann et al. developed a system for grading intervertebral disc degeneration ranging from Grade I to Grade V. Degenerative end-plate changes are also classified into six types depending on the presence of modic changes. Degenerative changes of the spinal joints are traditionally thought to be the results of insults such as mechanical or metabolic injury [1,2]. The disease entity encompasses a spectrum of structural anomalies of the spine including spondylosis, intervertebral disc disorder, and spinal stenosis that share similar symptoms of intolerable back pain with occasional radiation to the lower limbs due to nerve root compression. Previous studies have identified a number of risk factors for spinal joint degenerative changes, including obesity, diabetes, tobacco dependency, genetic factors, spinal fractures, and aging [2]. Although age is thought to be the most important risk factor associated with degenerative spinal changes [2,4], structural foot deformities (e.g., *pes planus*) have also been shown to be an independent risk factor contributing to spinal degeneration [5]. 

Hallux valgus, which is a common deformity around the first metatarsophalangeal joint, involves impaired rotation, abnormal angulation, and lateral deviation of the big toe [6]. The severity of deformity can be assessed by examining the hallux vulgus angle and the first–second intermetatarsal angle. Conventionally, a deformity was categorized into four types based on the hallux valgus angle, the intermetatarsal angle, and the distal metatarsal articular angle based on anteroposterior view of a radiograph [7]. 

As in degenerative spinal diseases, age plays an important role in hallux valgus; a previous study demonstrated a higher prevalence of hallux valgus in people aged over 65 years than in those aged 18–65 years (35.7% vs. 23%, respectively) [8]. Consistently, hallux valgus has been found to be associated with age-related functional disability such as reduced plantar tactile sensitivity [9], impaired balance [10], and the risk of falling [11]. The weakness of the transverse head adductor halluces muscle may result in collapse of the transverse arch, leading to inadequate impact absorption and causing forefoot pain in patients with hallux valgus [12]. In addition, the increased incidence of pes planus in adolescents with hallux valgus has been shown to be associated with recurrent knee-joint pain, lower-back pain (possibly caused by lower-limb rotation), and compensatory posture changes between the spine and lower extremities [12,13,14,15]. One study has shown that spinal deformity, lower-extremity alignment, and joint range of motion are highly associated with hallux valgus, especially among dancers due to changes in hip internal rotation and ankle plantar flexion [16]. However, there have been limited studies investigating the development of degenerative spinal diseases in relation to hallux valgus. 

In this study, we hypothesized that hallux valgus is a possible risk factor for degenerative spinal diseases. Using the National Health Insurance Registration Database (NHIRD) of Taiwan, we conducted a national population-based study to test the hypothesis that hallux valgus may be a risk factor for degenerative spinal changes.

## 2. Materials and Methods

### 2.1. Data Sources

The longitudinal NHIRD, which is overseen by the Health and Welfare Data Science Center (HWDC) in Taiwan [17], contains information on two million beneficiaries that were randomly sampled from the entire population registry for the year 2000. The database includes all outpatient and inpatient medical claims, including diagnoses based on the International Classification of Disease (ninth revision, Clinical Modification, ICD-9-CM), sex, age, medications, surgical operations, medical interventions and procedures, and fees from 2000 to 2015. To ensure privacy for the participants, the identification numbers were scrambled. The study was approved by the Institutional Review Board (IRB) of Chung Shan Medical University Hospital (approval number: CS1-20201).

### 2.2. Study Design

This was a retrospective cohort study involving participants who were newly diagnosed with hallux valgus (ICD-9-CM code 735.0) from 1 January 2001 to 31 December 2014. To ensure accuracy of the diagnoses, we included patients with at least two outpatient visits or at least one hospitalization. The index date was defined as the first date of the diagnosis of hallux valgus. To confirm new onset of the disease, we excluded those with a diagnosis of spinal degeneration (ICD-9-CM 721, 722, 723, and 724) before the index date. Non-hallux valgus subjects referred to those never diagnosed with hallux valgus from 2000 to 2015. 

The outcome variable was defined as the diagnosis of a symptomatic degenerative spinal disease, namely, spondylosis (ICD-9-CM 721), intervertebral disc disorder (ICD-9-CM 722), or spinal stenosis (ICD-9-CM 723 and 724), after the index date with at least two outpatient visits or at least one hospitalization. Both groups were followed until the onset of spinal degeneration, death, or 31 December 2015. Our sensitivity analysis also evaluated the outcomes using a washout period of 1 and 2 years for lowering the detection bias.

### 2.3. Covariates and Matching

As possible covariates, age and sex were analyzed along with medical comorbidities, such as hypertension (ICD-9-CM 401–405), hyperlipidemia (ICD-9-CM 272.0–272.4), chronic liver disease (ICD-9-CM 571), chronic kidney disease (ICD-9-CM 585), diabetes (ICD-9-CM 250), chronic obstructive pulmonary disease (ICD-9-CM 491, 492, 496), rheumatoid arthritis (ICD-9-CM 714.0), ankylosing spondylitis (ICD-9-CM code 720.0), heart failure (ICD-9-CM 428), hyperthyroidism (ICD-9-CM 242), and cancer (ICD-9-CM 140–208). Diagnoses that were considered to change the axial loading of body weight were also included, namely, lower-limb fracture (ICD-9-CM 820–829) and osteoarthritis (ICD-9-CM 715). 

First, a 1:4 matching by age and sex was used to provide an index date for the non-hallux valgus subjects that had the same starting point. Propensity score matching at a 1:1 ratio was then performed on the basis of age, sex, and comorbidities between the two groups. The propensity score was estimated using logistic regression analysis with greedy nearest neighbor. The binary variables were the hallux valgus and non-hallux valgus groups. Propensity score matching nearly balanced the heterogeneity of sex, age, and comorbidities between the two groups.

### 2.4. Statistical Analysis

Mean values are expressed as the mean ± standard deviation (SD). The hallux valgus and non-hallux valgus groups were compared using the chi-square test, Fisher’s exact test, or independent *t*-test, where appropriate. The relative risk (RR) and 95% confidence intervals (CIs) were calculated using a Poisson regression model. Kaplan-Meier analysis was used to calculate the cumulative incidence of degenerative spinal changes between the two groups. The log-rank test was used to test statistical significance. To determine the independent risk of the hallux valgus group, a multivariate Cox proportional hazard model was used to estimate the hazard ratio and adjusted hazard ratio (aHR) with 95% confidence intervals (CIs). The power analysis was based on a hazard ratio of 1.75, an alpha error of 0.05, an overall event probability of 0.323, and the total sample size of 2000. The statistical power was greater than 0.99. Statistical analysis was performed using SAS version 9.4 (SAS Institute Inc., Cary, NC, USA).

## 3. Results

We identified 3948 patients who were newly diagnosed with hallux valgus (ICD-9-CM 735.0) via outpatient visits or admissions from 2001 to 2014. After 1:4 age and sex matching, we identified 1023 hallux valgus and 4092 non-hallux valgus subjects. After further propensity score matching, we eventually included 1000 patients with and 1000 patients without hallux valgus (Figure A1). The demographic and clinical characteristics of the two study cohorts are presented in Table 1. The mean age of the hallux valgus group and that of the non-hallux valgus group after propensity score matching were 36.71 ± 17.31 and 36.47 ± 17.52, respectively. The majority of the subjects were females (74.4% in the hallux valgus group and 75.3% in the non-hallux vulgus group). Before propensity score matching, the hallux valgus group had a higher proportion of patients with hyperlipidemia, rheumatoid arthritis, and osteoarthritis than the non-hallux group. After propensity score matching, age, sex, and comorbidities were not significantly different between the two groups.

The incidence densities of spinal degeneration were 73.10 and 42.63 per 1000 person-years in the hallux valgus and non-hallux valgus groups, respectively. The relative risk for spinal degeneration associated with hallux valgus was 1.72 (95% CI 1.46–2.01) (Table 2). Kaplan-Meier survival analysis showed a significantly higher cumulative incidence of spinal degeneration in the hallux valgus group (log-rank *p* < 0.001) (Figure 1) than that in the non-hallux group. Compared with the non-hallux valgus group, patients with hallux valgus had an increased risk of spinal degeneration (aHR 1.75, 95% CI 1.50–2.05). Further sensitivity analysis still demonstrated increased relative risks for spinal degeneration associated with hallux valgus of 1.61 (95% CI 1.35–1.93) and 1.64 (95% CI 1.34–2.02) with 1 and 2 year washout periods, respectively (Table A3).

Regarding other significant risk factors for spinal degeneration including older age, chronic liver disease, heart failure, and osteoarthritis (Table 3), subgroup analysis showed a significantly higher risk for spinal degeneration after stratification by age and sex in the hallux valgus group than that in the non-hallux valgus group (Table 4). In addition, individuals in the hallux valgus group with a history of lower-limb fracture may be at increased risk of spinal degenerative change. However, there was no significant gender difference in the hallux valgus group (aHRs of 1.70 and 1.79 for females and males, respectively). There was also no notable trend of an increased risk of developing degenerative spinal change with increasing age in the hallux valgus group (aHRs: 1.90, 1.55, 1.84, and 1.78 for individuals aged <20, 20–39, 40–64, and ≥65 years, respectively) (Table 4).

Sub-outcome analysis revealed significantly higher risks for spondylosis (aHR 2.01, 95% CI 1.55–2.61), intervertebral disorder (aHR 2.27, 95% CI 1.62–3.17), and spinal stenosis (aHR 1.24, 95% CI 1.47–1.76) (Table A1) in the hallux valgus group compared to the non-hallux valgus group. Moreover, those in the hallus valgus group older than 40 years had a higher risk of lumbar stenosis than that of cervical stenosis (Table A1).

To further evaluate the effect of the surgical intervention, our result (Table A2) demonstrated an increased risk of spinal degenerative change for those in the hallux valgus group without surgical intervention (aHR = 1.95, 95% CI = 1.66–2.99, *p* < 0.001).

## 4. Discussion

This longitudinal population study with up to 14 years follow-up is the first to demonstrate an association between hallux valgus and an increased risk of developing degenerative spinal diseases. On the basis of our study, hallux valgus was associated with a 1.7-fold increased risk of degenerative spinal changes. Furthermore, we found that the risk for intervertebral disc disorder was the highest (aHR 2.27) among all the degenerative spinal diseases associated with hallux valgus. 

Previous studies have demonstrated that intervertebral disc and facet joints sustain approximately 70% and 30% of axial compression loading, respectively, in the healthy population. Morphological changes of any spinal components, including the intervertebral discs, joints, and ligaments, can contribute to degenerative changes [1]. Causes of abnormal mechanical axial pressure, such as trauma, osteoporosis, genetic background, and age, have been shown to be risk factors for degenerative changes of the spine [1,18,19,20]. In addition to abnormal mechanical axial pressure, systemic medical conditions such as diabetes mellitus have been found to promote the degeneration of intervertebral discs [21].

A recent literature review revealed an association between changes in foot structure and anterior knee or lower-back pain [22]. In view of the paucity of studies focusing on the direct relationship between the structural alterations of such a distal part of extremity (first metatarsal) and spinal degenerative disorders, the present large population-based longitudinal study divided patients into those with newly diagnosed hallux valgus and those without hallux valgus for evaluating the risk of developing degenerative spinal changes after propensity score matching. After age- and sex-stratified analysis, we found that patients with hallux valgus had a 1.75-fold increased risk of developing degenerative spinal diseases compared to those without. In addition, we found that gender and age may not be significant predictors of hallux valgus and the development of degenerative spinal diseases.

Hallux valgus, a common foot deformity in the modern world, often causes foot pain and functional disability. One study showed that hallux valgus is not an individual issue of the first ray, but a long-term progressive malfunction of the foot impacting the whole kinematics chain of the lower limb [23]. A number of risk factors have been reported, including age, genetic predisposition, and occupation. The improper footwear fit in length is also considered an extrinsic predisposing risk factor for developing hallux valgus [24] The pooled prevalence of hallux valgus was estimated to be 23% in adults aged 18–65 years, and 35.7% in people over 65 years of age [8]. Typically, abductor–adductor muscle imbalance is considered to be a major cause of hallux valgus [25], and some specific angles have been defined to quantify and evaluate the severity of the deformity, such as the hallux valgus angle (hallux valgus A: angle between the proximal phalanx of the hallux and the shaft axis of the first metatarsal) and the intermetatarsal angle (IMA: angle between the first and second metatarsal) [8,26]. A larger angle represents an increased likeliness of the occurrence of degenerative changes and progressive subluxation of the first metatarsophalangeal (MTP), potentially leading to an impaired balance and an increased risk of falling in older individuals [10,26,27]. In addition, restricted ankle dorsiflexion and reduced rearfoot supination have been reported in patients with hallux valgus [28]. Such restrictions in motion may predispose to an increase in valgus forces on the hallux for which the foot may compensate through external rotation [29]. 

Older people suffering from hallux valgus tend to have a decreased stride length and velocity when walking on an irregular surface [28]. In addition, biomechanically, hallux valgus may cause lateral shifting of the center of pressure and cause internal rotation of the hip, thereby potentially contributing to medial knee osteoarthritis. Asymmetry in hip rotation may eventually disrupt the kinetic chain from the foot to the back, leading to lower-back pain [30,31,32]. 

The association between hallux valgus and spinal degenerative changes may be explained from a biomechanical perspective. Hallux valgus is known to induce pronation of the foot and cause a change in native mechanical alignment, which is accompanied by an unbalanced shifting of axial loading (e.g., lumbar lordosis) as a compensatory mechanism [33,34]. A long-term increase in lumbar lordosis (hyperlordosis) may subsequently overload the spinal joints as supported by the result of a previous study that demonstrated a significant correlation between lordosis and degenerative spinal joint disease [35].

The major strength of this study is the use of a national-level dataset with a 14 year follow-up period. Because almost all residents are covered and enrolled in the Taiwanese insurance, this has benefits for the prediction of national trends. Moreover, the diagnoses of hallux valgus and degenerative spinal change were made by specialists. Additionally, we only recruited patients with the diagnosis of hallux valgus at least twice during outpatient visits or at least one hospitalization with the diagnosis to minimize misdiagnosis. Furthermore, to reduce selection bias, we performed multivariate modeling analysis to address potential confounding variables. We also observed the washout period up to 2 years for lowering the detection bias. 

Nevertheless, there were some limitations to this population-based case-control study. First, the number of individuals diagnosed with hallux valgus may have been underestimated because some patients may be asymptomatic in the absence of a trigger (e.g., shoe friction to the bunion). Furthermore, because our data source was confined to Taiwan, further studies from other countries are needed to extrapolate our findings to other populations. Second, the severity of the condition cannot be investigated using the ICD-9-CM code (735.0) without information about the angles for assessing clinical severity [8,26,29,36]. In addition, because we could not differentiate unilateral from bilateral hallux valgus, potential differences between unilateral and bilateral involvement in the degree of unbalanced shifting of axial loading via the distal limb to the central trunk remain unclear. Third, although genetic predispositions to hallux valgus and degenerative spinal changes have been reported [18,29,37], we did not evaluate the family background or conduct genomic research to investigate the possible impacts on our study outcomes. 

Fourth, despite the known association between many autoimmune diseases and spinal disorders, laboratory data, such as the erythrocyte sedimentation rate, C-reactive protein, rheumatoid factor, and uric acid levels, were not available for a detailed study; thus, the possible correlation between hallux valgus and autoimmune disease [38] warrants further studies for evaluation. Fifth, information on lifestyles, nutritional status, and occupation that may affect the occurrence of hallux valgus and spine degenerative disorders could not be retrieved from the dataset. Sixth, although our finding may be explained biomechanically, further anatomical and biomechanical studies are needed to elucidate this connection.

## 5. Conclusions

In conclusion, we found that, for the Taiwanese population, hallux valgus was associated with a 1.7-fold increased risk of degenerative spinal changes in a population-based cohort study. Further investigations are warranted to support our finding.

## Figures and Tables

**Figure 1 ijerph-20-01152-f001:**
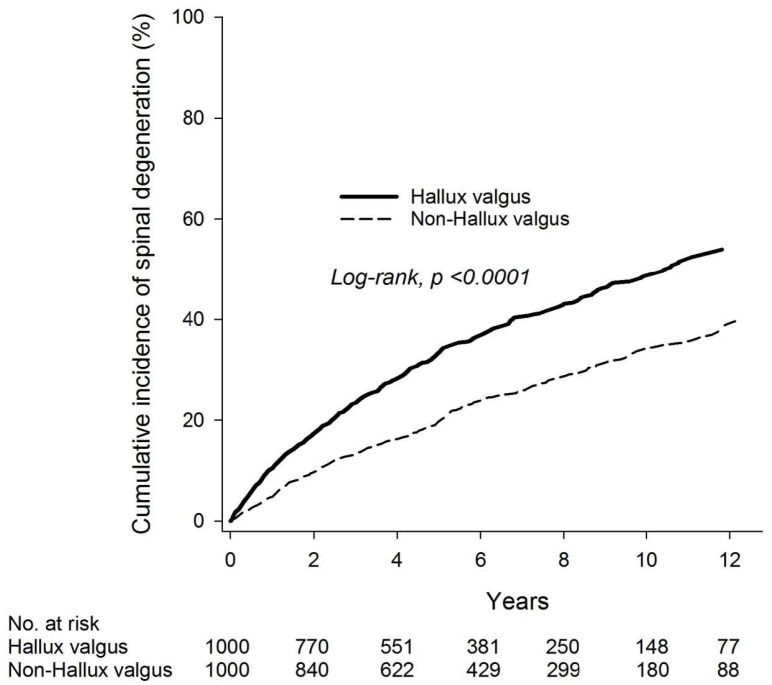
Kaplan–Meier analysis for incidence of spinal degeneration.

**Table 1 ijerph-20-01152-t001:** Demographic characteristics of the hallux valgus and non-hallux valgus groups.

	Before PSM Matching					After PSM Matching				
	Non-Hallux Valgus (*n* = 4092)		Hallux Valgus (*n* = 1023)			Non-Hallux Valgus (*n* = 1000)		Hallux Valgus (*n* = 1000)		
	*n*	%	*n*	%	*p*-Value	*n*	%	*n*	%	*p*-Value
Age (years)					1					0.9914
<20	684	16.7	171	16.7		176	17.6	171	17.1	
20–39	1660	40.6	415	40.6		408	40.8	411	41.1	
40–64	1448	35.4	362	35.4		348	34.8	351	35.1	
≥65	300	7.3	75	7.3		68	6.8	67	6.7	
Mean ± SD	37.11 ± 17.51		37.11 ± 17.52		1	36.47 ± 17.52		36.71 ± 17.31		0.757
Sex					1					0.6428
Female	3040	74.3	760	74.3		753	75.3	744	74.4	
Male	1052	25.7	263	25.7		247	24.7	256	25.6	
Hypertension	316	7.7	72	7.0	0.460	73	7.3	68	6.8	0.6623
Hyperlipidemia	105	2.6	45	4.4	0.0019	42	4.2	40	4.0	0.8216
Chronic liver disease	77	1.9	22	2.2	0.577	24	2.4	22	2.2	0.7654
Diabetes	159	3.9	18	1.8	0.0009	18	1.8	18	1.8	1.0000
COPD	43	1.1	14	1.4	0.3866	9	0.9	10	1.0	0.8177
Rheumatoid arthritis	5	0.1	16	1.6	<0.0001	4	0.4	5	0.5	1 ^¶^
Heart failure	12	0.3	6	0.6	0.157	3	0.3	3	0.3	1 ^¶^
Hyperthyroidism	9	0.2	6	0.6	0.0958 ^¶^	4	0.4	6	0.6	0.5261
Cancer	51	1.2	17	1.7	0.2994	16	1.6	14	1.4	0.7129
Lower-limb fracture	26	0.6	11	1.1	0.1376	13	1.3	11	1.1	0.6813
Osteoarthritis	59	1.4	48	4.7	<0.0001	34	3.4	33	3.3	0.9011

^¶^ Fisher’s exact test. COPD, chronic obstructive pulmonary disease; PSM, propensity score matching.

**Table 2 ijerph-20-01152-t002:** Poisson regression of relative risk of hallux valgus and non-hallux valgus groups.

	Non-Hallux Valgus	Hallux Valgus
*n*	1000	1000
Person-years	5982	5349
No. of spinal degeneration	255	391
ID (95% CI)	42.63 (37.70–48.19)	73.10 (66.20–80.72)
Relative risk (95% CI)	Reference	1.72 (1.46–2.01)

ID, incidence density (per 1000 person-years); CI, confidence interval.

**Table 3 ijerph-20-01152-t003:** Cox proportional hazard model analysis for risk of spinal degeneration.

	Univariate		Multivariate ^†^	
	HR (95% CI)	*p*-Value	HR (95% CI)	*p*-Value
Group				
Non-hallux valgus	Reference		Reference	
Hallux valgus	1.69 (1.45–1.98)	<0.0001	1.75 (1.50–2.05)	<0.0001
Age				
<20	Reference		Reference	
20–39	1.99 (1.47–2.69)	<0.0001	1.99 (1.47–2.69)	<0.0001
40–64	3.72 (2.77–4.99)	<0.0001	3.61 (2.68–4.86)	<0.0001
≥65	3.90 (2.70–5.62)	<0.0001	3.43 (2.28–5.14)	<0.0001
Sex				
Female	Reference		Reference	
Male	0.83 (0.69–1.00)	0.046	0.86 (0.71–1.04)	0.111
Hypertension	1.89 (1.47–2.44)	<0.0001	1.25 (0.92–1.70)	0.158
Hyperlipidemia	1.69 (1.21–2.37)	0.002	1.07 (0.74–1.53)	0.734
Chronic liver disease	2.23 (1.53–3.23)	<0.0001	1.69 (1.15–2.50)	0.008
Diabetes	1.14 (0.67–1.94)	0.626	0.63 (0.35–1.12)	0.115
COPD	2.15 (1.19–3.91)	0.012	1.14 (0.57–2.30)	0.709
Rheumatoid arthritis	0.35 (0.05–2.45)	0.287	0.26 (0.04–1.88)	0.184
Heart failure	6.86 (2.84–16.58)	<0.0001	4.07 (1.44–11.47)	0.008
Hyperthyroidism	0.50 (0.13–2.01)	0.331	0.40 (0.10–1.63)	0.203
Cancer	1.20 (0.64–2.23)	0.575	0.79 (0.42–1.50)	0.475
Lower-limb fracture	1.27 (0.68–2.37)	0.453	1.25 (0.66–2.39)	0.495
Osteoarthritis	1.79 (1.26–2.55)	0.001	1.48 (1.03–2.13)	0.033

COPD, Chronic obstructive pulmonary disease. ^†^ Adjusted for age, sex, hypertension, hyperlipidemia, chronic liver disease, diabetes, COPD, rheumatoid arthritis, heart failure, hyperthyroidism, cancer, lower-limb fracture, and osteoarthritis.

**Table 4 ijerph-20-01152-t004:** Subgroup analysis for risk of spinal degeneration among the hallux valgus and non-hallux valgus groups.

	Non-Hallux Valgus		Hallux Valgus		Multivariate ^†^	
	*n*	No. of Spinal Degenerations	*n*	No. of Spinal Degenerations	HR ^†^ (95% CI)	*p*-Value
Age						
<20	176	19	171	33	1.90 (1.08–3.34)	0.026
20–39	408	90	411	131	1.55 (1.19–2.03)	0.001
40–64	348	121	351	188	1.84 (1.46–2.31)	<0.0001
≥65	68	25	67	39	1.78 (1.07–2.94)	0.026
*p* for interaction =						0.7842
Sex						
Female	753	201	744	302	1.70 (1.42–2.04)	<0.0001
Male	247	54	256	89	1.79 (1.28–2.51)	<0.001
*p* for interaction =						0.9188
Lower-limb fracture						
No	987	NA	989	383	1.69 (1.44–1.98)	<0.0001
Yes	13	NA	11	8	7.50 (1.43–39.37)	0.017
*p* for interaction =						0.0333
Osteoarthritis						
No	966	240	967	373	1.71 (1.46–2.01)	<0.0001
Yes	34	15	33	18	2.28 (1.12–4.66)	0.024
*p* for interaction =						0.0937

^†^ Adjusted for age and sex.

## Data Availability

Not applicable.

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
