# Peer review of "Association of Hallux Valgus with Degenerative Spinal Diseases: A Population-Based Cohort Study"

_ijerph, 2023, doi:10.3390/ijerph20021152_

Round 1

Reviewer 1 Report

Well done on the research, and overall, I feel your manuscript was concise well written. I have a few suggestions: 

1. Line 52-53 you make the statement that degenerative changes have no universal definition. I feel this is a bold statement. There are a number of ways to grade spinal degeneration. For example, the article below compares  possible examples of a way to grade degenerative disc disease. 

Quint U, Wilke HJ. Grading of degenerative disk disease and functional impairment: imaging versus patho-anatomical findings. Eur Spine J. 2008 Dec;17(12):1705-13. doi: 10.1007/s00586-008-0787-6. Epub 2008 Oct 7. PMID: 18839226; PMCID: PMC2587674.

2.  Your outcome variable was defined as the diagnosis of a degenerative spinal disease (line 100) after the index date. My suggestion would be to refer to this as a symptomatic degenerative spinal disease. A large portion of individuals have spinal degenerative change (see article below), but do not experience symptoms that might bring them in for care. My assumption is that the two outpatient visits or hospitalization (lines 102-103) were for symptoms associated with diagnosed degenerative changes. I wold explain this as a limitation. 

Brinjikji W, Luetmer PH, Comstock B, Bresnahan BW, Chen LE, Deyo RA, Halabi S, Turner JA, Avins AL, James K, Wald JT, Kallmes DF, Jarvik JG. Systematic literature review of imaging features of spinal degeneration in asymptomatic populations. AJNR Am J Neuroradiol. 2015 Apr;36(4):811-6. doi: 10.3174/ajnr.A4173. Epub 2014 Nov 27. PMID: 25430861; PMCID: PMC4464797.

Author Response

Comments and Suggestions for Authors

Well done on the research, and overall, I feel your manuscript was concise well written. I have a few suggestions: 

  1. Line 52-53 you make the statement that degenerative changes have no universal definition. I feel this is a bold statement. There are a number of ways to grade spinal degeneration. For example, the article below compares possible examples of a way to grade degenerative disc disease. 

Quint U, Wilke HJ. Grading of degenerative disk disease and functional impairment: imaging versus patho-anatomical findings. Eur Spine J. 2008 Dec;17(12):1705-13. doi: 10.1007/s00586-008-0787-6. Epub 2008 Oct 7. PMID: 18839226; PMCID: PMC2587674.

Answer: Thank you for providing an article that clarifies the grading of degenerative disc disease, so I have revised the article accordingly to remove the statement "there is no universal definition".

  1. Your outcome variable was defined as the diagnosis of a degenerative spinal disease (line 100) after the index date. My suggestion would be to refer to this as a symptomatic degenerative spinal disease. A large portion of individuals have spinal degenerative change (see article below), but do not experience symptoms that might bring them in for care. My assumption is that the two outpatient visits or hospitalization (lines 102-103) were for symptoms associated with diagnosed degenerative changes. I wold explain this as a limitation. 

Brinjikji W, Luetmer PH, Comstock B, Bresnahan BW, Chen LE, Deyo RA, Halabi S, Turner JA, Avins AL, James K, Wald JT, Kallmes DF, Jarvik JG. Systematic literature review of imaging features of spinal degeneration in asymptomatic populations. AJNR Am J Neuroradiol. 2015 Apr;36(4):811-6. doi: 10.3174/ajnr.A4173. Epub 2014 Nov 27. PMID: 25430861; PMCID: PMC4464797.

Answer: Thank you for the great advice. I changed line 100 to add "symptomatic" (line 120) before degenerative spine disease.

Reviewer 2 Report

I assume from the text that only diagnoses with proven structural disorders in the spine were included - 3 were included (spondylosis, intervertebral disc disorder, spinal stenosis).
If there were other degenerative changes in the spine that were not included in the selection, please clarify the selection criteria.The great variability of possible clinical pictures makes it difficult to establish specific relationships, but the discussion details the limits of the study.

Author Response

I assume from the text that only diagnoses with proven structural disorders in the spine were included - 3 were included (spondylosis, intervertebral disc disorder, spinal stenosis).
If there were other degenerative changes in the spine that were not included in the selection, please clarify the selection criteria. The great variability of possible clinical pictures makes it difficult to establish specific relationships, but the discussion details the limits of the study.

Answer:

Thank you for the great advice .The article below titled Degenerative disease of the spine has mentioned spondylosis, intervertebral disc disorder, spinal stenosis as a cause of continuous changes as a response to physiological axial load. Thus I chose these three diagnoses as most possible related to mechanical shifting due to hallux valgus.

DOI: doi:10.1016/j.nic.2007.01.002

Reviewer 3 Report

The researchers aimed to determine by conducting a population-based study in Taiwan the hypothesis that hallux valgus may be a risk factor for degenerative changes of the spine.

The manuscript has errors and requires corrections by the investigators, the comments are shown below:

The work requires an English review by an expert; please attach certification of the English review.

Because the title indicates "degenerative spinal diseases" and only "Spondylosis" is indicated in the keywords because they prioritize this lesion with respect to others in the degenerative disorders of the spine.

In the introduction, they must give a detailed and complete description of the study population; incidence; classification; diagnosis; current treatments; physical, psychological, social and economic problems associated with the injury, as recommended by the official  degenerative spinal diseases (an important aspect of the rationale and importance of the study) in both the introduction and the discussion.

The introduction is very scarce in content and does not contextualize the problem. The number of references used is very scarce, only 5 references are used to address and justify a problem as important as the one they are investigating, association of spinal disorders with hallus valgus, rewrite your introduction.

In the introduction there is no context/foundations to explain the reasons and the scientific basis of the research that the article deals with, you try to unify all the "degenerative spinal diseases" but no classification is provided, nor is there any discussion about the pathologies that are going to be the subject of the study.

Attach a checklist for cohort studies following the STROBE model. Follow STROBE guidelines.

Enter in the methodology section, a subsection called "study design" to present the key elements of the study design. Remember that this guideline is focused on cross-sectional, case-control, and cohort designs, so you should provide the characteristics and justification of the design you have used.

Enter in the methodology section a subsection called "context" (framework or conditions under which the study was conducted, the relevant places and dates, including recruitment, exposure, follow-up and data collection periods).

Enter in the methodology section, subsection "data sources/measurements": for each variable of interest, provide the data sources and details of the measurement methods used. If more than one group, specify the comparability of measurement processes.

10º Enter in the methodology section, subsection "Biases" specify all measures taken to manage potential biases.

11º Enter in the methodology section, subsection "Sample size" explain how the sample size was determined. This is very relevant for the validity of the results, being important to consider the confidence level and the statistical power used.

12º Enter in the methodology section, subsection "Quantitative variables" explain how the quantitative variables were treated in the analysis. If variables were grouped, the reason for this should be specified.

13º In the results section, in reference to the participants, describe the number of participants in each phase of the study; the target group, who met the selection criteria and those who were finally selected, followed up and analyzed, as well as the losses at each stage, especially in the follow-up. A good way to summarize this information is with a flow chart.

14º In the results section, in cohort studies describe the number of positive events and provide unadjusted and, if appropriate, confounder-adjusted estimates and their precision (e.g., 95% confidence intervals). Specify the confounding factors for which you adjust and the reasons for including them. If categorizing continuous variables, describe the limits of the intervals. If relevant, along with estimates of relative risk, add estimates of absolute risk for a relevant time period.

15º Discussions should cover the key findings of the study: discuss any previous research related to the topic to place the novelty of the discovery in the appropriate context, discuss possible shortcomings and limitations in its interpretations, discuss its integration into the current understanding of the problem and how. This advances current views, speculates on the future direction of research, and freely postulates theories that could be tested in the future, completed, and reformulated. The discussion should be rewritten to present serious errors.

16º In the discussion section, these four key points have not been considered and addressed in detail and in depth by the authors: "key results" (summarize the main results of the study, always in accordance with the stated objectives); "limitations" (discuss the limitations of the study, taking into account possible sources of bias or imprecision in obtaining the results. Discuss the direction and magnitude of possible biases, as well as the effect they would have on the results obtained). These are several: the study on a very specific population and region (difficulty in generalizing the results); the dispersion of pathologies covered by the concept of "degenerative spinal diseases", among others; "interpretation" (provide a prudent global interpretation of the results considering objectives, limitations, multiplicity of analyses, results of similar studies and other relevant empirical evidence) and "generalization" (discuss the external validity of the results, that is, the possibility of generalizing them to other conditions).

17º The conclusions should be reformulated and expanded according to the results obtained, since you have done it in a specific population of a specific country and this statement is overstated "In conclusion, we found that hallux valgus was associated with a 1.7-fold increased risk of degenerative spinal changes in a population-based cohort study".

18º The bibliography does not comply with the guidelines for authors of the journal, the number of references is very scarce and deficient when it comes to justifying the introduction and discussion. A high number of references are more than 10 years old, which indicates a research and justification not in accordance with the current evidence, see references 2,3,6,7,8,9,9,10,11,12,13,18,20,21,23,24,25,27,32.

Author Response

The manuscript has errors and requires corrections by the investigators, the comments are shown below:

 The work requires an English review by an expert; please attach certification of the English review.

Response: Thanks for kindly reminder, I’ll attach certification of the English review.

 Because the title indicates "degenerative spinal diseases" and only "Spondylosis" is indicated in the keywords because they prioritize this lesion with respect to others in the degenerative disorders of the spine.

Response: Thanks for kindly reminder, I deleted one of the key word “degenerative spinal diseases” and I retained spondylosis in the keywords as your suggestion.

 In the introduction, they must give a detailed and complete description of the study population; incidence; classification; diagnosis; current treatments; physical, psychological, social and economic problems associated with the injury, as recommended by the official degenerative spinal diseases (an important aspect of the rationale and importance of the study) in both the introduction and the discussion.

Response: Thanks for kindly reminder, I rewrote the introduction as your suggestion.

 The introduction is very scarce in content and does not contextualize the problem. The number of references used is very scarce, only 5 references are used to address and justify a problem as important as the one they are investigating, association of spinal disorders with hallus valgus, rewrite your introduction.

Response: Thanks for kindly reminder, I rewrote the introduction and added two new references which has mentioned types (classification) of hallux vulgus and one of the references has proofed spinal deformity, lower limb alignment and joint range of motion are strongly related with hallux valgus in dancer’s population

5º In the introduction there is no context/foundations to explain the reasons and the scientific basis of the research that the article deals with, you try to unify all the "degenerative spinal diseases" but no classification is provided, nor is there any discussion about the pathologies that are going to be the subject of the study.

Response: Thanks for kindly reminder, I rewrote the introduction and provided grading, classification about the degenerative spinal changes of the intervertebral disc degeneration, end plate changes (from line 53-59). The severity of HV deformity were also added (from line 73-77)

6º Attach a checklist for cohort studies following the STROBE model. Follow STROBE guidelines.

Response: Thank you for your suggestion. I attached a checklist for cohort studies following the STROBE model.

 Enter in the methodology section, a subsection called "study design" to present the key elements of the study design. Remember that this guideline is focused on cross-sectional, case-control, and cohort designs, so you should provide the characteristics and justification of the design you have used.

Response: Thank you for your suggestion. We had presented the "study design" subsection in line 91-99. This study is focused on cohort design. The hallux valgus and non-hallux valgus groups were followed until the onset of spinal degeneration, death, or December 31, 2015.

8º Enter in the methodology section a subsection called "context" (framework or conditions under which the study was conducted, the relevant places and dates, including recruitment, exposure, follow-up and data collection periods).

Response: Thank you for your suggestion. We had presented the context description in line 91-99. The study group was newly diagnosed with hallux valgus (ICD-9-CM code 735.0) from January 1, 2001 to December 31, 2014. Non-hallux valgus subjects referred to those never diagnosed with hallux valgus from 2000 to 2015. Both groups were followed until the onset of spinal degeneration, death, or December 31, 2015.

 Enter in the methodology section, subsection "data sources/measurements": for each variable of interest, provide the data sources and details of the measurement methods used. If more than one group, specify the comparability of measurement processes.

Response: Thank you for your suggestion. The data source/measurements were presented in line 81-115. The data sources was from the two million beneficiaries National Health Insurance Registration Database (NHIRD) in Taiwan. The measurement of variable of interest was diagnosed by the International Classification of Disease (ninth revision, Clinical Modification, ICD-9-CM). The study group was newly diagnosed with hallux valgus (ICD-9-CM code 735.0) from January 1, 2001 to December 31, 2014. Non-hallux valgus subjects referred to those never diagnosed with hallux valgus from 2000 to 2015.

10º Enter in the methodology section, subsection "Biases" specify all measures taken to manage potential biases.

Response: Thank you for your suggestion. The efforts to reduce the potential biases were presented in line 116-122. We used the propensity score matching that was estimated using logistic regression analysis with greedy nearest neighbor. By matching the propensity score, it could balance the heterogeneity of the two groups among age, sex, and comorbidities.

11º Enter in the methodology section, subsection "Sample size" explain how the sample size was determined. This is very relevant for the validity of the results, being important to consider the confidence level and the statistical power used.

Response: Thank you for your suggestion. We added this for explanation as below in line 132-134: The power analysis was based on Hazard ratio of 1.75, alpha error of 0.05, overall event probability of 0.323, and the total sample size of 2000. The statistical power was greater than 0.99.

12º Enter in the methodology section, subsection "Quantitative variables" explain how the quantitative variables were treated in the analysis. If variables were grouped, the reason for this should be specified.

Response: Thank you for your suggestion. We had presented the context description in line 91-99. The study group was newly diagnosed with hallux valgus (ICD-9-CM code 735.0) from January 1, 2001 to December 31, 2014. Non-hallux valgus subjects referred to those never diagnosed with hallux valgus from 2000 to 2015.

13º In the results section, in reference to the participants, describe the number of participants in each phase of the study; the target group, who met the selection criteria and those who were finally selected, followed up and analyzed, as well as the losses at each stage, especially in the follow-up. A good way to summarize this information is with a flow chart.

Response: Thank you for your suggestion. We described the number of participants in each phase of the study in line 137-141. A flow chart was presented in Figure A1.

14º In the results section, in cohort studies describe the number of positive events and provide unadjusted and, if appropriate, confounder-adjusted estimates and their precision (e.g., 95% confidence intervals). Specify the confounding factors for which you adjust and the reasons for including them. If categorizing continuous variables, describe the limits of the intervals. If relevant, along with estimates of relative risk, add estimates of absolute risk for a relevant time period.

Response: Thank you for your suggestion. We described the number of spinal degeneration in table 2. The incidence densities of spinal degeneration was described in line 150-152. Unadjusted and confounder-adjusted model were estimated in table 3. The Hazard ratio for risk of spinal degeneration was described in line 155-159.

15º Discussions should cover the key findings of the study: discuss any previous research related to the topic to place the novelty of the discovery in the appropriate context, discuss possible shortcomings and limitations in its interpretations, discuss its integration into the current understanding of the problem and how. This advances current views, speculates on the future direction of research, and freely postulates theories that could be tested in the future, completed, and reformulated. The discussion should be rewritten to present serious errors.

Response: Thank for your kindly reminder. I rewrote the discussion as your suggestions.

16º In the discussion section, these four key points have not been considered and addressed in detail and in depth by the authors: "key results" (summarize the main results of the study, always in accordance with the stated objectives); "limitations" (discuss the limitations of the study, taking into account possible sources of bias or imprecision in obtaining the results. Discuss the direction and magnitude of possible biases, as well as the effect they would have on the results obtained). These are several: the study on a very specific population and region (difficulty in generalizing the results); the dispersion of pathologies covered by the concept of "degenerative spinal diseases", among others; "interpretation" (provide a prudent global interpretation of the results considering objectives, limitations, multiplicity of analyses, results of similar studies and other relevant empirical evidence) and "generalization" (discuss the external validity of the results, that is, the possibility of generalizing them to other conditions).

Response: Thank for your kindly reminder. I rewrote the discussion as your suggestions. (The added sentences were presented in line 216-217, 282-284)

17º The conclusions should be reformulated and expanded according to the results obtained, since you have done it in a specific population of a specific country and this statement is overstated "In conclusion, we found that hallux valgus was associated with a 1.7-fold increased risk of degenerative spinal changes in a population-based cohort study".

Response: Thank for your kindly reminder. I rewrote the conclusion as your suggestions. (The renewed sentence was presented in line 303)

18º The bibliography does not comply with the guidelines for authors of the journal, the number of references is very scarce and deficient when it comes to justifying the introduction and discussion. A high number of references are more than 10 years old, which indicates a research and justification not in accordance with the current evidence, see references 2,3,6,7,8,9,9,10,11,12,13,18,20,21,23,24,25,27,32.

Response: dear reviewers, thank for your kindly reminder. I added two new references less than 10 years old to support my research.

STROBE checklist

Title and abstract

1a. Indicate the study’s design with a commonly used term in the title or the abstract

Reply: the study’s design with a commonly used term in the title or the abstract was presented in line 1-3 and 22-41

1b. Provide in the abstract an informative and balanced summary of what was done and what was found

Reply: An informative and balanced summary was presented in line 39-40

Introduction

  1. Background/rationale. Explain the scientific background and rationale for the investigation being reported

Reply: the scientific background and rationale for the investigation being reported was presented in line 44-91

  1. Objectives. State specific objectives, including any prespecified hypotheses

Reply: prespecified hypotheses was presented in line 92-95

Methods

  1. Study design. Present key elements of study design early in the paper

Reply: The study design was presented in line 112-126.

  1. Setting. Describe the setting, locations, and relevant dates, including periods of recruitment, exposure, follow-up, and data collection

Reply: The Setting was presented in line 81-99.

  1. Participants. Give the eligibility criteria, and the sources and methods of selection of participants

Reply: The Participants was presented in line 92-99.

  1. Variables. Clearly define all outcomes, exposures, predictors, potential confounders, and effect modifiers. Give diagnostic criteria, if applicable

Reply: The Variables was presented in line 92-104, 107-114.

  1. Data source/measurements. For each variable of interest, give sources of data and details of methods of assessment (measurement). Describe comparability of assessment methods if there is more than one group

Reply: The data source/measurements were presented in line 100-110.

  1. Describe any efforts to address potential sources of bias

Reply: The efforts to reduce the potential sources of bias were presented in line 120-126.

  1. Study size. Explain how the study size was arrived at

Reply: We added the power analysis in line 132-134.

  1. Explain how quantitative variables were handled in the analyses. If applicable, describe which groupings were chosen and why

Reply: The sentence was presented in line 127-143.

  1. Statistical methods.

Reply: The sentence was presented in line 127-156.

Results

  1. (a) Report numbers of individuals at each stage of study—eg numbers potentially eligible, examined for eligibility, confirmed eligible, included in the study, completing follow-up, and analysed. (b) Give reasons for non-participation at each stage(c) Consider use of a flow diagram

Reply: A flow diagram was showed in figure A1.

  1. Descriptive data. (a) Give characteristics of study participants (eg demographic, clinical, social) and information on exposures and potential confounders (b) Indicate number of participants with missing data for each variable of interest (c) Cohort study—Summarise follow-up time (eg average and total amount)

Reply: The result was showed in table 1, and table 3.

  1. Outcome data. Report numbers of outcome events or summary measures

Reply: The result was showed in table 2.

  1. Main results. (a) Report the numbers of individuals at each stage of the study—eg numbers potentially eligible, examined for eligibility, confirmed eligible, included in the study, completing follow-up, and analysed (b) Give reasons for non-participation at each stage (c) Consider use of a flow diagram

Reply: The result was showed in line 137-159 and figure A1.

  1. Other analyses. Report other analyses done—eg analyses of subgroups and interactions, and sensitivity analyses

Reply: We performed the analyses in table 4 and table A3.

Discussion

  1. Key results. Summarise key results with reference to study objectives

Reply: key results was presented in line 158-198

  1. Limitations. Discuss limitations of the study, taking into account sources of potential bias or imprecision. Discuss both direction and magnitude of any potential bias

Reply: The limitations were presented in line 279-301

  1. Interpretation. Give a cautious overall interpretation of results considering objectives, limitations, multiplicity of analyses, results from similar studies, and other relevant evidence

Reply: The interpretation were presented in line 303-305

  1. Generalisability. Discuss the generalisability (external validity) of the study results

Reply: The Generalizability were presented in line 214-278

Other information

  1. Funding. Give the source of funding and the role of the funders for the present study and, if applicable, for the original study on which the present article is based

Reply: The Funding were presented in line 311 (This research received no external funding.)

Round 2

Reviewer 3 Report

We congratulate the authors for the partial improvements made to the manuscript.

Although some of the issues raised remain unresolved, I will enumerate some of them:

1º We do not understand whether the work focuses on spondylosis or on all degenerative diseases of the spine, this aspect has to be clarified the title indicates one thing and the content of the manuscript explains the opposite.

2º The STROBE guidelines are not 100% complied with, some sections are well defined in subsections, others are not.

3º Although it was indicated to the authors an in-depth revision of the introduction and discussion, this has not been done.

4º The Tables summarize their findings but this is not described in the manuscript.

5º The number of old citations, which do not provide relevant and current information, is high.

6º The bibliographic citation guidelines established by the journal are not complied with (review the document).

7º The tables summarize the findings but this is not described in the manuscript.

It would be interesting to review the recommendations indicated above and solve them. We have neither the STROBE checklist nor certification of English review by an expert.
